# Socioecological Factors Associated with an Urban Exercise Prescription Program for Under-Resourced Women: A Mixed Methods Community-Engaged Research Project

**DOI:** 10.3390/ijerph18168726

**Published:** 2021-08-18

**Authors:** Sarah M. Camhi, Gifty Debordes-Jackson, Julianna Andrews, Julie Wright, Ana Cristina Lindsay, Philip J. Troped, Laura L. Hayman

**Affiliations:** 1Department of Kinesiology, University of San Francisco, 2300 Fulton Street, San Francisco, CA 94117, USA; julie.wright@umb.edu (J.W.); ana.lindsay@umb.edu (A.C.L.); Phil.troped@umb.edu (P.J.T.); 2Department of Exercise and Health Sciences, University of Massachusetts Boston, 100 Morrissey Blvd, Boston, MA 02125, USA; 3Department of Sociology, University of Massachusetts Boston, 100 Morrissey Blvd, Boston, MA 02125, USA; gifty.debordes-ja001@umb.edu (G.D.-J.); Julianna.Andrews001@umb.edu (J.A.); 4Department of Nursing, University of Massachusetts Boston, 100 Morrissey Blvd, Boston, MA 02125, USA; laura.hayman@umb.edu

**Keywords:** mixed methods, physical activity, exercise prescription, under-resourced, chronic disease, U.S., urban, socioecological model

## Abstract

One strategy to promote physical activity (PA) is for health care providers to give exercise prescriptions (ExRx) that refer to community-based facilities. However, facilitators and barriers specific to urban programs in the US for under-resourced women are unknown. Thus the purpose of this formative research was to explore ExRx barriers and facilitators specific to US under-resourced women to inform future intervention targets and strategies. This mixed-methods community-engaged research was conducted in partnership with an urban women’s only wellness center that exchanged ExRx for free access (1–3 months). Qualitative semi-structured interviews and validated quantitative questionnaires (SF-12, International Physical Activity Questionnaire, Physical Activity Self-Efficacy, Physical Activity Stage of Change, and Barriers to Physical Activity, Social Support for Exercise, and Confusion, Hubbub, and Order Scale) were administered by phone and guided by the socio-ecological model. ExRx utilization was defined as number visits/week divided by membership duration. Means and percentages were compared between ≥1 visit/week vs. <1 visit/week with *t*-tests and chi-square, respectively. Women (*n* = 30) were 74% Black, 21–78 years of age, 50% had ≤ high school diploma, and 69% had household incomes ≤45,000/year. Women with ≥1 visit/week (*n* = 10; 33%) reported more education and higher daily activity, motivation, number of family CVD risk factors and family history of dyslipidemia compared with <1 visit/week. Facilitators among women with ≥1 visit/week were “readiness” and “right timing” for ExRx utilization. Barriers among women with <1 visit/week (*n* = 20; 67%) were “mismatched expectations” and “competing priorities”. Common themes among all women were “sense of community” and “ease of location”. ExRx utilization at an US urban wellness center may be dependent on a combination of multi-level factors including motivation, confidence, peer support, location and ease of access in under-sourced women. Additional resources may be needed to address mental and/or physical health status in additional to physical activity specific programming.

## 1. Introduction

Obesity is increasing in the United States (US) with projections that 40–50% of adults will be obese by 2030 [1]. Obese adults are more likely to have abnormal cardiovascular disease (CVD) risk factors [2], which increases their risk for CVD mortality and morbidity [3]. It is well documented that increasing physical activity (PA) can improve obesity and CVD risk factors [4]. Despite the benefits of PA, 40.5% of US adults do not meet the federal PA guidelines [5].

The health care provider has an important influence in promoting regular PA. The health care office is the most common place to receive health services [6] and health care providers have been shown to positively influence PA [7]. Several policy initiatives have been developed to encourage health providers to more regularly promote PA in their patients. The American College of Sports Medicine’s Exercise is Medicine initiative utilizes various PA promotion strategies in the health care setting such as advice/counseling, technology, and exercise prescriptions (ExRx) [8]. ExRx can provide a patient with a connection to community-based facilities that can provide accessible and safe PA programming and/or education with qualified staff [9,10]. In addition, the US Preventive Services Task Force (USPSTF) recommends that health care providers refer patients with CVD risk factors to PA counseling [11]. Approximately 37% of adults are eligible for PA counseling according to the USPSTF criteria [12]. Over half of the eligible adults also do not meet the PA guidelines [12], indicating a large number of individuals who could potentially benefit from ExRx programs.

Despite these policy initiatives for health care providers to promote PA, ExRx resources, even when prescribed, are under-utilized. Research shows 1/3 of patients who receive an ExRx do not utilize it at all; 12–42% complete the recommended length of programming [13], with attrition occurring throughout the program [14]. Even lower ExRx utilization and participation are found in women [14,15] and are particularly low among women with chronic disease risk factors [16]. Barriers to ExRx adherence include lack of self-efficacy, social support, and access to exercise facilities for patients in the United Kingdom and European countries [13,17], though results vary by country due to differences in policies and health care systems [18]. Several studies have identified the need to further identify factors that will increase engagement with ExRx in the US [12,19], and whether these strategies would be successful in lower income countries [20].

The recent release of the World Health Organization’s updated PA guidelines highlight the gaps in knowledge for promotion of PA in special populations including people with chronic disease and disabilities [21]. The need to “consider their national context and factors such as culture, ethnic diversity, existing social norms” (pg. 1460) [21] is especially important to reduce health inequities that exist in these groups for both low PA and increased risk for chronic disease. Studies have recognized the importance of health care providers recognizing personal situations and context to evaluate whether people are open to change (i.e., motivation), though it was also recognized that the evidence from low-income and racial/ethnic groups was limited [22]. Understudied groups such as urban and under-resourced US populations have unique needs to consider for PA interventions such as cost, social and physical environmental factors [23] as well as family values [24]. Therefore, the purpose of this research was to explore individual, interpersonal and environmental barriers and facilitators associated with ExRx adherence in under-resourced women in an urban environment as formative research to inform future interventions.

## 2. Materials and Methods

This mixed methods community-engaged research was conducted in partnership with a community-based non-profit 501c(3) public charity women’s community fitness and wellness center located in Dorchester, MA, USA. The center provides fitness and wellness related services with subsidized membership rates to women at all income levels, with a sliding scale of membership rates determined by income. At the time this research was conducted in 2018, it served approximately 1800 active members with 93% identifying as Black. Dorchester is a community whose obesity rates are 11% higher than the surrounding Boston metropolitan area (33% vs. 22%) [25].

The ExRx program was initiated in 2008, and allows patients with diabetes, obesity, hypertension and mood disorders to exchange ExRx from participating local community health centers for complimentary access and membership to the wellness center. Memberships were either 1- or 3-months depending on the specific community health center agreement and/or program. The main goal of the ExRx program was to reduce barriers related to cost of accessing a wellness facility. All members were given access to the wellness facility equipment, classes, personnel and resources. Beyond free access, the ExRx program did not have any specific standardized physical activity programming for the participants. Each member’s recommendations were based on conversations with their health care provider and/or wellness center staff that were appropriate for their individual goals. Previous published research from the program’s first year of operation has shown that participant characteristics such as being Black and having more comorbidities correlated to utilizing the wellness center [16]. After 10 years of operation, it was unknown which aspects of the ExRx program were successful and where improvements may be needed.

In order to better understand the ExRx program needs, the researchers worked with both the wellness facility personnel (Director, staff) and the affiliated local community health center staff (health care providers, administrative staff) to identify information related to the ExRx that would be helpful and meaningful to evaluate the current methods and procedures. Staff and personnel were interviewed by the PI to learn more about how the program worked, its history and anecdotal evidence relating to successes and challenges. Questions for the ExRx program participants were developed based on the mutually agreed upon needs identified by the researchers, the wellness center staff and community health center staff. To better understand the complexity of the factors that might influence the utilization of the ExRx and to encourage a deeper and richer assessment of the women’s experience and identify potential barriers and facilitators of ExRx program, a mixed-methods approach was utilized to gather both qualitative and quantitative information.

A recent systematic review on ExRx acknowledged a general lack of theoretical frameworks and/or assessment related to behavior change theory and called for more research in this area [26]. Research approaches using a socio-ecological model are recommended to better understand ExRx engagement and utilization [16], with an emphasis on groups with low participation such as in low income women [17] and/or Black women [18,27]. A socio-ecological model framework was utilized to guide the methodologies in which multiple levels of influence on exercise were examined [28]. The socio-ecological framework was used to develop the interview protocol and questions at the individual (barriers, challenges, and knowledge), interpersonal (social environment and social support), environmental (access to the wellness center and/or e-health), organizational (healthcare system) and policy levels. Utilizing the socio-ecological model allowed researchers to investigate the relationship, interactions and communication between the individual and organizations, and how it informs activation and utilization of the ExRx.

A convenience sample of women with ExRx memberships was recruited from the wellness center. The sample was identified via the wellness center’s database of members who joined with an ExRx from a health care provider. Patients (*n* = 250) were invited to participate via postcards, letters, emails, flyers, phone calls and/or in person at the wellness center. Women were eligible if they were 18 years and older, active members of the wellness center, and received an ExRx between 1 January 2017 and 31 December 2018. Participants were excluded if they were pregnant or breastfeeding or had any physical limitations that would preclude them from participating in exercise or PA. All protocols and procedures were approved by the Institutional Review Board at the University of Massachusetts Boston (protocol ID: 2018025). All participants provided informed consent prior to participation.

After a 5-min screening call to determine eligibility, researchers conducted two telephone interviews with participants with quantitative questionnaires occurring first and qualitative interviews occurring second. The total time for telephone interviews was approximately 60–90 min. All interviews were conducted via phone after consultation with wellness center staff and health care center administration. Due to the high cancellation and no-show rate for medical and health-related appointments in-person, we chose to conduct all sessions via phone to enhance participation. All participants received USD 50 gift cards for participation.

### 2.1. Quantitative Methods

The quantitative questionnaires phone interviews were conducted by two trained research assistants in which their interview skills were pre-tested before formal data collection to ensure neutrality. Self-report questionnaires were used to assess individual-level factors including age, race/ethnicity, socio-economic status, education, health status (hypertension, diabetes and CVD), family health history (obesity, hypertension, diabetes and CVD), height, weight, smoking status, alcohol intake, sitting/day, usual activity (i.e., sitting, standing, lifting loads) and television and computer use. Validated questionnaires were used to assess other individual level factors and included: Perceived mental and/or physical health (SF-12) [29], PA (International Physical Activity Questionnaire) [30], PA self-efficacy [31], stage of change [32] and barriers to being physically active [33]. Interpersonal factors included social support for exercise/PA [33,34] and family stress (Confusion, Hubbub and Order Scale, CHAOS [35]). Home address was used to create three objective measures of neighborhood environment that included walkability (Walkscore [36,37]), bikability (BikeScore) and access to public transportation (TransitScore). Distance in miles (walking and driving) of patient’s home from wellness center was estimated using google maps.

#### Quantitative Data and Statistical Analysis

Wellness center utilization was defined as the total number of times a participant visited the center after their initial sign-up, divided by the total number of weeks of the participant’s membership. Utilization was measured via a membership card scanned into a computer software system which occurred upon entry to gain access to the wellness center at each visit post sign-up. Utilization was split into two categories to reflect regular and consistent usage ≥1 wellness center visits/week versus irregular or inconsistent usage defined as <1 wellness center visit/week. Individual, interpersonal and environmental variables were confirmed to have normal distributions and values were compared between ≥1 and <1 visits/week using *t*-tests for continuous variables and chi-square analysis for categorical variables. All statistical analyses were performed using SAS software v.9.4 (Cary, NC, USA) where significance was set at *p* < 0.05.

### 2.2. Qualitative Methods

All telephone qualitative interviews were conducted by the PI who was trained in qualitative methodologies. The interview guide was developed, revised and modified in a collaborative effort between wellness center staff and the research team of investigators before formal data collection which included two additional researchers with qualitative experience (JW and ACL). Telephone interviews were audio recorded using Audacity^®^ software (https://audacityteam.org/, accessed on 10 February 2019). One-on-one interviews asked about participants’ experiences receiving the ExRx and using the wellness center during their membership. Questions focused on identifying barriers and/or challenges related to the participant’s experience with the referral process from the health care provider, ExRx activation (i.e., exchanging the ExRx for a membership), ExRx utilization (i.e., visiting the wellness center) and/or barriers or difficulties they may have experienced while attempting to engage in PA/exercise. Interviews were conducted with patients until saturation of themes was found and/or whether themes identified were consistent with the available qualitative literature related to working with underserved and racial/ethnic groups in urban settings for PA programming [27,38].

All interviews were transcribed by a professional transcriptionist and de-identified to ensure privacy. All transcripts were coded by two research assistants (GDJ and JA). In developing the codebook, research assistants coded segments of the interview transcripts and assigned codes. Once research assistants agreed on the selected segments and applied codes, the information was transferred to the qualitative software Dedoose Version 8.0.35. Dedoose is a web application for managing and presenting qualitative and mixed method research data (Los Angeles, CA, USA: SocioCultural Research Consultants, LLC www.dedoose.com, accessed on 10 February 2019). The research assistants utilized two phases while coding the data. In phase one of coding and analysis, the two assistants coded the same transcripts and met to discuss the themes and codes that would inform the codebook.

The research assistants engaged in an initial coding utilizing the “lumping” approach in which they categorized each new excerpt with a word or short phrase [39]. This approach enabled investigators to develop a codebook based on the interview excerpts. The research assistants reviewed each excerpt again to ensure a shared understanding of the codes and consistency in coding. During the second stage of coding, the research assistants applied the established codes to new transcripts. Research assistants reviewed these excerpts again to ensure consistency in code application. Once research assistants reached a consistent shared understanding of the codes, they met with the PI to review the codebook and other potential codes that may have been overlooked. Data collection, coding and data analysis were iterative [40]. As researchers continued to code and analyze interview data, a codebook was developed to provide definition of each code and excerpt(s) that reflects the code. The main themes and codes included in the present study focus specifically on the participant’s barriers and/or facilitators related to ExRx utilization.

## 3. Results

Approximately 250 women were contacted from the wellness center which resulted in *n* = 48 women screened to determine eligibility. Of the total screened, 37 were eligible to participate and 30 women (81% of eligible sample) completed both quantitative and qualitative interviews; 26 women (70% of eligible sample) completed the quantitative interviews only. All women who completed quantitative interviews were included in the quantitative analyses (*n* = 30), and all women who completed the qualitative interviews were included in the qualitative analyses (*n* = 26).

Women (*n* = 30) identified as 74% non-Hispanic Black, with a mean age 42.5 ± 13.4 years and a mean BMI of 35.5 ± 7.2 kg/m^2^. Approximately 50% had a high school diploma or less, 34% were currently employed full time and 69% had a household income <$45,000/year. Women reported marital status as 60% single and had on average of 2.1 ± 1.3 children and 1.3 ± 1.4 dependents (Table 1).

Women received their ExRx from seven different health centers in and around the Boston area. ExRx were most often made by physicians (*n* = 19; 63%), but also by nurses (*n* = 5; 17%), a nutritionist (*n* = 1), a psychiatrist (*n* = 1) and unidentified or unknown type of health care provider (*n* = 4). The self-reported diagnosis related to receiving an ExRx was 53% obesity, 20% musculoskeletal disorders, 20% mental health/mood disorders, 13% hypertension and 10% diabetes; however, 30% of the women reported more than 1 health issue. Women reported that they learned about the ExRx program from their physician (*n* = 17; 56%); word of mouth from friends/family (*n* = 6; 20%); from the wellness center (*n* = 2; 7%) or unknown or don’t remember (*n* = 5; 16%). Approximately 50% of the women had been a member of a fitness or wellness center before receiving an ExRx. While most women had a 3-month membership (*n* = 19; 63%), some had 1 month (*n* = 5; 17%), or other (i.e, extended membership of additional months due to extenuating circumstances, *n* = 2; 6%).

The average number of visits/week during the membership was 0.7 ± 1.1 (range 0–5.3 visits/week). Of the 30 participants, 67% (*n* = 20) had <1 visit/week, and 33% (*n* = 10) with ≥1 visit/week. Of the women who were classified <1 visit/week (*n* = 10), four women had 0 visits after their initial membership sign-up.

After the free trial membership was over, participants were given the option to extend their membership for a discounted rate. Overall, 65% (*n* = 17) reported continuing their membership which was equally distributed among <1 visit/week (*n* = 8) and ≥1 visit/week utilizers (*n* = 9). Among those who did not continue their membership, the majority of women were <1 visit/week (*n* = 6; 75%) compared to only 2 ≥1 visit/week (25%).

Table 2 presents the quantitative data for individual, interpersonal and environmental factors by utilization group (≥1 visit/week vs. <1 visit/week). Facilitators and barriers identified from the qualitative interviews are summarized in Figure 1 by the individual, interpersonal and environmental levels from the socio-ecological model.

### 3.1. Individual Level

#### 3.1.1. Quantitative Results

Quantitative surveys revealed that those who were more educated had higher ExRx utilization (*p* = 0.02) (Table 2). There were no other differences between utilization groups for sociodemographic variables at the individual level for age, race, income, employment, or health status (number of CVD risk factors, BMI, CVD, diabetes, hypertension, dyslipidemia, musculoskeletal issues or medication usage).

For health and lifestyle behaviors, quantitative surveys results showed that those whose usual daily activities involved lifting and carrying loads were more likely to have greater ExRx utilization (*p* = 0.03) compared to those who sit/stand. Although women with ≥1 visit/week reported levels of walking, moderate, vigorous and total PA approximately two times higher than women with <1 visit/week (total PA MET*mins: <1 visit/week: 3184 ± 3402; ≥1 visit/week: 5352 ± 5609); these differences were not statistically significant (*p* = 0.22). Women with ≥1 visit/week also reported a lower amount of sedentary behavior (sitting, TV or computer use), but levels were not significantly different from women with <1 visit/week. There were no significant differences between utilization groups in smoking status or alcohol intake (Table 2).

Results from the quantitative surveys showed that self-perceived mental and physical health were not significantly different between utilization groups in the quantitative surveys, however, a score on the SF-12 of ≤45 is considered below average and indicates impairment. For mental health, the <1 visit/week group reported a SF-12 score that was ~45 (some impairment with 1 standard deviation below population mean), while the ≥1 visit/week group indicated a SF-12 score of ~41 (impairment with ~2 standard deviations below population mean) [41]. Qualitative interviews did reveal that women in the <1 visit/week group reported that mental and physical illness (i.e., depression/social anxiety/arthritis) were reasons that kept them from visiting the wellness center. In the quantitative survey, among those who had <1 visit/week, low motivation was significantly higher compared to those who had ≥1 visit/week (*p* = 0.01; Table 2). Other barriers to PA in the quantitative survey were not significantly different between the utilization groups included lack of time, social influence, lack of energy, fear of injury, lack of skill and lack of resources (Table 2).

#### 3.1.2. Qualitative Results

A prominent theme among the <1 visit/week group during the qualitative interviews was a barrier of “competing issues and/or priorities” (Figure 1). Women reported dependent care (children and/or other family members) and/or financial issues as common issues. One quote from an unemployed mother of 3 children said: “I really really appreciated that the program existed and that I could use that to get back into an exercise routine. Unfortunately, I just had a lot going on at the time…” (Female, 27 years).

During the qualitative interviews, women with ≥1 visit/week indicated that it was the “right time” for them to incorporate PA into their lives (Figure 1). One participant was a student who was balancing school and a full-time job, said: “I was on a break and I got more time than I usually have. So I ended up being able to go more than I would have any other time of the year” (Female, 25 years). Thus, timing of when the ExRx was given in the context of someone’s other personal responsibilities may be an important factor to consider. In support of these findings, the quantitative surveys revealed a higher percentage of women with ≥1 visit/week were in the action stage of change regarding PA (67% in women with ≥1 visit/week compared to 50% in women with <1 visit/week 50%), although the difference between the groups was not significant (*p* = 0.40) (Table 2).

Women with ≥1 visit/week also reported during the qualitative interviews that they felt a “readiness to change” (Figure 1). One participant who received her ExRx for obesity management stated “I was ready, your head has to be ready in anything that you do…I just knew it was something I had to do and I was ready to get it done. So I went and got it done… I don’t like exercising. I don’t like the gym. But I know in order to achieve my goal, which was to lose weight, I had to go. I had to move” (Female, 54 years of age). This quote demonstrates increased motivation and ability to incorporate PA which is in agreement with the trends from the quantitative data for self-efficacy. Women with ≥1 visit/week did trend towards higher scores in self efficacy compared with women with <1 visit/week, however, the difference between the groups was not significant (*p* = 0.08).

### 3.2. Interpersonal Level

#### 3.2.1. Quantitative Results

Women with ≥1 visit/week reported more family history of dyslipidemia (*p* = 0.04) and the number of cardiovascular disease risk factors compared to women with <1 visit/week (*p* = 0.04; Table 2). This may indicate a particular facilitator related to family history and/or health that may motivate a participant to utilize ExRx programming. There were no other significant differences between the utilization groups concerning other interpersonal factors as measured by the quantitative questionnaires such as marital status, number of children or dependents, people in the household or CHAOS score (family and home stress) or support for PA from family (*p* = 0.99) and/or friends (*p* = 0.15) (Table 2).

#### 3.2.2. Qualitative Results

Despite the lack of differences in quantitative data for family and friend social support for PA, there was a clear theme from the interviews of lack of staff support regarding the services at the center for women with <1 visit/week. Women who had <1 visit/week reported having “mismatched expectations” regarding the services that the wellness center provided (Figure 1). They wanted more support, education and services to inform and guide them on PA and other lifestyle and health issues. One participant stated “I think I just needed the information, training, encouragement and more like, a buddy to help you get started or a trainer to help you get started. Somebody to break the ice…it’s a scary thing to go into something…you know you can’t do these goals by yourself because you have no idea where to start…but there’s really no one to actually show or help you out in the weights” (Female, 54 years). Women expressed that their needs were not addressed concerning their health, and they wanted additional programming and services as part of the membership beyond access to the wellness center. One woman with diabetes stated: “they know that women of color, one of the biggest things is being a diabetic. And they should have more information concerning that…” (Female, 61 years). Another woman who was given an ExRx to manage her obesity stated, “I think nutrition is like really key, because you can do all the exercises in the world, but if you’re not eating right…” (Female, 54 years). It is important to note that the experience of the initial visit to the wellness center varied across participants. Women may have received none, some or all of the following services: a facility tour, an orientation to the fitness equipment and/or personalized recommendations for classes/services. The women who received more time and attention at their initial visit when they were signing up reported more positive experiences relating to connecting with a particular staff member, class/service and/or other peer member. Thus, while some of these services were available at the center (i.e., personal trainer and/or nutrition counseling), these services were inconsistently introduced during their orientation and/or tour during their first visit. While the initial personal training and/or nutrition counseling session was complimentary, further visits/sessions may have involved fees and thus may not have been accessible for some participants. In conclusion, the first impression of the member had a lasting effect on their overall experience and participation in the ExRx program.

A prominent theme that emerged as a facilitator among all participants, regardless of utilization, was the “sense of community” (Figure 1). As one participant ≥1 visit/week stated, “It’s basically like a big family…We talk, and its friendly and you know, everyone’s trying to get in shape” (Female, 22 years, ≥1 visit/week). Women also felt “acceptance” when visiting a women’s only center (Figure 1). Women’s reasons for why they enjoyed visiting the wellness center often included words and phrases such as “comfortable”, “no judgement”, “no competition” and “able to focus”. Women reported feeling “acceptance” as evidenced by this quote: “But if I had to make a choice, I would go rather go…where I know the woman are going to look more like me, as far as being a little full” (Female, 51 years, <1 visit/week). Women also reported feeling motivated by seeing others like them: “Okay well that person looks like she’s my height, my size, trying to do the same thing I’m doing. She can do it, I can do it…I see the same faces all the time, because it’s a smaller gym…it motivates me too…” (Female, 25 years, ≥1 visit/week).

### 3.3. Environmental Level

#### 3.3.1. Quantitative Results

At the environmental level, women with in both utilization groups had similar walkability, bikability and access to transit within their neighborhoods as estimated by WalkScore, BikeScore and TransitScore, respectively. The estimated average walkability of the participants’ neighborhood was considered “Very Walkable: Most errands can be accomplished on foot” [42]. While there was no significant difference in distance to the wellness center from their home between <1 visit/week (1.5 ± 2.1 miles) and ≥1 visit/week (2.1 ± 1.7 miles) (Table 2), it is important to note that the mean miles indicated that the wellness center was located within their local neighborhood.

#### 3.3.2. Qualitative Results

The qualitative data indicate a theme related to “location” for the women regardless of utilization that acted as a facilitator to ExRx utilization (Figure 1). As stated by this participant who lived less than a 5-min walk from the wellness center: “I started, I think in February. It was cold, we had snow. It was within walking distance. I was able to just walk to the gym, get done, come home” (Female, 54 years, ≥1 visit/week). The location of the wellness center within her immediate neighborhood allowing her the ability to walk, helped lessen other PA barriers such as weather. Other important themes from the qualitative interviews that related to location included the “ease of access” related to availability of parking, and that its location was close to other services related to health and wellness (i.e., cooking classes, healthy grocery store) (Figure 1).

## 4. Discussion

This community-engaged, mixed methods study explored barriers to and facilitators to utilization of an urban ExRx program from health care providers in under-resourced women. In summary, facilitators of ≥1 visit/week among under-resourced women in the urban ExRx program included higher education, higher usual daily activity, family history of cardiovascular risk factors, and a “readiness for change” and “right timing”. Barriers associated with <1 visit/week included lower motivation/willpower, and “mismatched expectations” and “competing priorities”. For all women, regardless of utilization, important factors regarding their positive experience at the center related to the “ease of access”, “location”, “sense of community” and “acceptance”.

Access to and location of the wellness center was an important factor to ExRx utilization in women in the current study. Another study has shown that women located in urban and affluent areas are more likely to use local PA facilities in Canada, although this association was not found in men [43]. Location of a wellness center to be within walking distance of home appeared to ease the burden of needing a car, navigating public transportation and/or parking specifically in lower socioeconomic ethnic minorities in the Netherlands [17]. Convenient location has also been shown in previous research to be an important indicator of participation in ExRx in older urban African American women in the US [44]. Our research extends these findings about the importance of easy access to facilities within the local neighborhood for urban under-resourced women in the US of younger ages. Further, our research shows that given that the wellness center location was also near other important health-related services (grocery and health education services), this made it an especially attractive and more frequented destination.

Previous studies have shown barriers in ExRx utilization to be related to cost in migrant and ethnic minority women living in low-income neighborhoods in the Netherlands [17]. In the current study the ExRx program was free of charge so cost was not prohibitive. However, the main barriers that our women reported for not utilizing the ExRx related more to “competing priorities” which was similar to UK studies which found social circumstances related to home, work and personal responsibilities to be a major barrier in men and women [45,46] and illness/injury specific to men and women with chronic diseases from the UK [47].

We did not observe any significant difference in participants personal health status for ExRx utilization in our study. Our study indicated that family history of chronic disease risk factors (i.e., dyslipidemia) that separated utilization groups <1 visit/week vs. ≥1 visit/week, and not personal health. Other studies have had mixed conclusions regarding health status. For example, in some studies, adherence varied by type of chronic disease: improved adherence in those with diabetes versus cardiovascular disease or obesity [18,48] or with increasing numbers of health conditions and/or medications [49]. In contrast, results from studies in older adults have been conflicting and shown that increasing or multiple health concerns have both negatively impacted adherence [50] and had no effect on adherence [44]. It is important to note that our participants represented a broad range of ages (21–78 years), reported multiple CVD risk factors, were predominately Black and Hispanic, and reported lower than average physical and mental health which may explain the difference found in results from previous studies.

Previous research has shown that for both men and women, the presence or lack of social support shaped their experiences as positive or negative, respectively [22,46,49,51]. One UK study found that lack of staff support and/or feeling ostracized was associated with poor participation for ExRx programs [46]. Increasing social support results in higher PA and adherence to ExRx programs [52]. Strong staff support has also been shown in previous research to be an important facilitator of ExRx adherence [22,46,51]. The women with <1 visit/week in the current study who reported “mismatched expectations” stated that they needed more supervision and education from staff, which is in agreement with previous research [22,46,49,51]. However, women in the current study also reported feeling motivated, “accepted” and supported by both peers and staff as evidenced by the “sense of community” theme. This theme of “community” was important to participants with both utilization groups. While it did not differentiate between the utilization groups, it is possible that other barriers noted (i.e., “mismatched expectations” and “competing priorities”) may have had a stronger influence on utilization in the <1 visit/week group.

Previous studies in ethnic minority women in the Netherlands including migrant Muslim and Dutch have shown that women have reported shame or embarrassment as a barrier to utilizing ExRx due to not feeling they were represented in size, shape or health in the facilities they were attending [17]. In contrast, our findings in US under-resourced urban women are different and show that women felt “acceptance” for their size and shape and motivation to see others who had similar health issues in the center. The different results may be due to differences in cultural contexts and norms between the US and Netherlands and/or racial/ethnic groups represented in the two studies. We also showed in our research that peer support and role-modeling from other women was important in this group, which is consistent with research which found that social comparisons were important for ExRx program success in the UK [46].

The women’s only aspect of the gym was an especially important feature for all women in the current study regardless of utilization. This comfort in attending women’s only facilities has also been shown with Muslim women in the United Kingdom [53]. In the US, research in PA has shown that women may feel embarrassed in mixed-sex groups and that migrant women were more comfortable being around other women [17].

Our qualitative results show that concepts relating to “readiness” and “right timing” were important factors among the urban under-resourced women who had ≥1 visit/week utilization. Swedish studies have found similar results that a balance in life and motivation are important predictors of success [54], whereby a lack of motivation is a predictor of ExRx non-adherence [55].

Strengths of the current study are that it included under-resourced women utilizing an urban ExRx program. The women were 74% Black which is a group that has higher prevalence of obesity and associated chronic conditions including diabetes and CVD [56,57]. Black women in particular are more likely to have low levels of PA [58], and thus are an important target for finding appropriate and effective strategies to increase PA. This study involved mixed methods community engaged design which allows for both a qualitative and quantitative approach to exploring possible facilitators and barriers associated with ExRx utilization. However, this methodology limits the sample size which can contribute to bias in our quantitative results. Despite the small sample size, our quantitative and qualitative results showed overall agreement and consistency with previous studies. However, results need to be replicated with larger sample sizes to confirm these findings. Further, there was a large age range represented in this sample. Given that age has been shown to influence barriers specific to physical activity [59], future studies are needed to explore results stratified by age. This research focused on women with obesity, mood disorders, and/or risk factors for diabetes and cardiovascular disease. Thus, these results may not be generalizable to other women with chronic diseases such as cancer. Since this research involved only one urban wellness center program in the Boston area, results might not be generalizable to other groups in other areas. Utilization was defined as accessing and visiting the wellness center and did not quantify actual type, time, duration or intensity of PA. It is possible that women may have accessed other wellness or fitness centers and/or PA programming in conjunction with access to this wellness facility and thus a limitation of the current study. The response rate for participation was low, and the sample may be biased towards people who had either strong positive or negative experiences associated with using the ExRx. This study focused on participants who accessed the wellness center in some capacity and we were not able to examine influences on women who received an ExRx from their health care provider but did not join the wellness center. Future studies are warranted to explore additional factors or characteristics of those who did not use or follow-up with the ExRx.

## 5. Conclusions

Our research suggests that the success of urban ExRx programs in US under-sourced women may be dependent on a combination of multi-level factors including motivation, confidence, peer support and location and ease of access of resources relative to home. Additional tailored resources may be needed to address other health issues such as women’s mental and/or physical health status, as well as programming and education specific to fitness and physical activity. Studies have confirmed that ExRx programs are more successful when the health care provider focuses on the personalized and tailored messaging and options [54,60]. Our results show that evaluating a person’s personal situation can have important implications for when, how and what to offer as part of the ExRx program. This information can aide in improving utilization of ExRx programs in urban settings for under-resourced women.

Future studies examining ExRx strategies should consider replication to confirm results and expand to other under-resourced groups. While the goal of this current research was to identify facilitators and barriers of ExRx utilization, future studies are also needed to explore whether ExRx can increase PA and/or mental and physical health in these populations.

## Figures and Tables

**Figure 1 ijerph-18-08726-f001:**
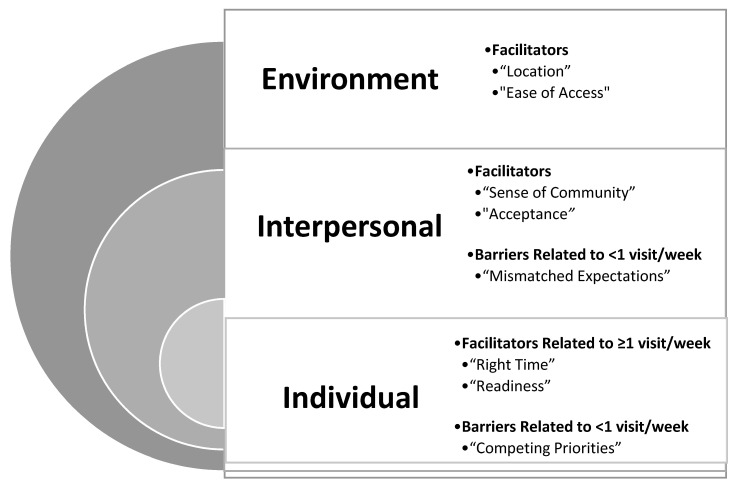
Multi-level qualitative barriers and facilitators associated with exercise prescription utilization.

**Table 1 ijerph-18-08726-t001:** Socio-demographics of women who received an exercise prescription.

	Total (*n* = 30)
**Age** (years)	42.5 ± 13.4
Range (years)	21–78
**Race/Ethnicity** *n* (%)	
Black (Hispanic/Non-Hispanic)	22 (74)
Hispanic	4 (13)
Mixed/Other	4 (12)
**Education** *n* (%)	
High school diploma or less	15 (50)
Some college	6 (20)
Associate degree/2 year college	3 (10)
College degree	6 (20)
**Employment** *n* (%)	
Full-time	10 (34)
Part time	7 (23)
Unemployed	12 (40)
Retired/other	1 (3)
**Marital Status** *n* (%)	
Single	18 (60)
Divorced	5 (17)
Married	5 (17)
Other	2 (6)
**Children** *n* (%)	
0	6 (20)
1–2	11 (37)
3–4	13 (43)
Number of Dependents	1.3 ± 1.4
**Household Income** *n* (%)	
<45,000/year	18 (69)
≥45,000/year	8 (31)
**People in Household**	3.5 ± 2.2

Note: Data are presented as means ± standard deviation for continuous variables and *n* (%) for categorical variables.

**Table 2 ijerph-18-08726-t002:** Comparison of individual, interpersonal and environmental factors by of exercise prescription utilization (*n* = 30).

	Utilization		*p*-Value
Definition	<1 Visit/Week	≥1 Visit/Week	
Individual Factors: Demographics and Health Status			
***n*** (% of total)	20 (67)	10 (33)	
**Age** (years)	42.2 ± 13.2	42.9 ± 14.3	0.9
Race: Black (Hispanic/Non-Hispanic) *n* (%)	15 (75)	7 (70)	0.72
Hispanic	3(15)	1 (10)	
Other (mixed race and other)	2 (10)	2 (20)	
**BMI** (kg/m^2^)	35.4 ± 7.9	35.7 ± 6.0	0.92
**Employment** (full or part time) *n* (%)	12 (60)	5(50)	0.6
**Education**: < high school *n* (%)	7 (35)	10 (80)	0.02 ^+^
**Household Income**: <45,000/year	14 (73)	4 (57)	0.42
**Health**: Self-perceived Mental Health (SF-12)	45.3 ± 13.5	41.9 ± 13.2 *	0.54
Self-perceived Physical Health (SF-12)	48.2 ± 7.5	53.3 ± 6.2	0.09
CVD Risk factors (#)	1.0 ± 1.0	0.6 ± 1.0	0.37
Diabetes *n* (%)	3 (15)	0 (0)	0.2
Hypertension *n* (%)	8 (40)	3 (30)	0.59
Musculoskeletal Disorder *n* (%)	6 (30)	2 (20)	0.56
Dyslipidemia *n* (%)	2 (10)	1 (10)	0.99
Cardiovascular Disease *n* (%)	0	0	-
Medication *n* (%)	15 (75)	6 (60)	0.40
**Individual Factors: Health Behaviors**			
**Sedentary Behavior**: TV (≤2 h/day)	13 (65)	7 (70)	0.78
Computer (≤2 h/day)	8 (40)	6 (60)	0.3
Sitting (mins/day)	268 ± 211	307 ± 193	0.65
**Usual Daily Activity**: Sits	5 (25)	1 (10)	
Stand/walks	9 (45)	1 (10)	0.03 ^+^
Lift loads	6 (30)	8 (80)	
**Physical Activity**: Walking (MET * mins)	766 ± 1041	1210 ± 1034	0.31
Moderate (MET * mins)	1036 ± 2180	2333 ± 3943	0.38
Vigorous (MET * mins)	1338 ± 2302	1809 ± 1936	0.60
Total (MET * mins)	3184 ± 3402	5352 ± 5609	0.22
**Smoking**: Current *n* (%)	2 (10)	1(10)	0.96
Former *n* (%)	9(45)	4(40)	
**Alcohol**: (drinks/week)	1.1 ± 2.9	0.6 ± 1.1	0.48
**Physical Activity Self-efficacy** (Range 1–5)	2.5 ± 0.8	3.1 ± 0.9	0.08
**Stage of Change**: Pre-action	10 (50)	3 (33)	0.40
Action *n* (%)	10 (50)	6 (67)	
**Barriers**: Lack of Time (Range 0–9) ^#^	3.0 ± 2.6	1.6 ± 2.3	0.18
Social Influence	3.9 ± 2.3	2.2 ± 2.4	0.09
Lack of Energy	4.3 ± 2.9	2.0 ± 2.9	0.07
Lack of Willpower/Motivation	6.0 ± 2.5 ^#^	2.8 ±2.8	0.007 ^+^
Fear of Injury	2.0 ± 2.4	1.2 ± 1.8	0.40
Lack of Skill	1.2 ± 1.3	0.9 ± 1.5	0.55
Lack of Resources	4.0 ± 2.9	2.2 ± 2.3	0.12
**Interpersonal and Social Factors**			
**Marital Status**: Married *n*%	4 (20)	1 (10)	0.48
**Children**	2.1 ± 1.1	2.2 ± 1.7	0.78
**Number of Dependents**	1.6 ± 1.5	0.8 ± 0.9	0.15
**People in Household**	3.2 ± 1.6	4.1 ± 3.0	0.39
**CHAOS**: Range 15–60	28.5 ± 11.1	27.3 ± 7.5	0.78
**Social Support for Physical** Activity Family Participation (Range 10–80)	20.4 ± 10.0	20.4 ± 8.9	0.99
Friend Participation	21.4 ± 9.4	16.6 ± 5.9	0.15
**Family Health History** CVD risk factors (#)	1.1 ± 1.1	2.0 ± 1.2	0.04 ^+^
Diabetes *n* (%)	7 (35)	4 (40)	0.79
Hypertension *n* (%)	9 (45)	8 (80)	0.07
Dyslipidemia *n* (%)	3 (15)	5 (50)	0.04 ^+^
CVD *n* (%)	2 (10)	3 (30)	0.17
**Environmental Factors**			
WalkScore (Range 0–100)	79.9 ± 13.2	78.6 ± 11.0	0.79
TransitScore (Range 0–100)	71.7 ± 6.4	70.2 ± 5.7	0.55
BikeScore (Range 0–100)	59.8 ± 9.0	63.3 ± 7.2	0.31
Walking distance to health center (miles)	1.5 ± 2.1	2.1 ± 1.7	0.58

**Key:** CHAOS: Confusion, Order and Hubbub Scale. Data are presented as means ± standard deviation for continuous variables and *n* (%) for categorical variables. * SF-12 scores for perceived health range 0–100; numerical score for mental or physical health <45 indicates impaired function and below average. ^#^ Scores ≥5 considered an important barrier to overcome. ^+^ *p*-values are statistically significant *p* < 0.05; continuous variables were tested via *t*-test; categorical variables were tested via chi-square. **Note:** Numbers vary for some variables since not all women answered all survey questions.

## Data Availability

Data are available upon reasonable request to the corresponding author.

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
