# Peer review of "Socioecological Factors Associated with an Urban Exercise Prescription Program for Under-Resourced Women: A Mixed Methods Community-Engaged Research Project"

_ijerph, 2021, doi:10.3390/ijerph18168726_

Round 1
Reviewer 1 Report
This study investigated in a group of obese women in an urban environment the impact of individual, interpersonal and environmental barriers and facilitators associated with adherence to exercise prescription. The exercise prescription was defined as mean number visits/week by membership duration (low < 1x and high > 1x visit/week). The conclusion was that exercise prescription use in an urban wellness center may depend on a combination of factors at various levels, including motivation, confidence, peer support, location, and ease of access in obese under-resourced women.
What is not known about barriers, facilities, exercise prescription and adherence in the population with cardiovascular risk factors. Mainly in obese people. This study did not show anything new, it is necessary to break paradigms, create new actions and effective intentions to reduce obesity.
Major comments:
There is insufficient detail describing the low < 1x and high > 1x visit/week;
This study has a type II error, the sample size is not large enough to detect a practical difference, when there really is a;
ExRx program needs more details (FITT = frequency, intensity, type, time)
Reviewer 2 Report
Abstract
Clearly present the objective of the study before the method.
Was the questionnaire used validated? Quote the instrument name
Introduction
There is no citation of the exercise protocol in countries outside the US-euporean axis, are there no studies? It is important to mention this, since in Central and South America, Asia and Africa, many countries have low-income women who could benefit from the protocol and its benefits
The last paragraph is too long
Methods
Start this section by pointing out the type of study before citing the location
Present the entire exercise program as well as its authors
Who performed the collection was trained? Were the same people for the quantitative and qualitative collection? Have you been trained for both collections? Have interviewers been pre-tested to ensure their skills?
Was the work approved by ethics committees? Submit the protocol approval number.
In the last paragraph, present the data analysis program, as well as the tests that were applied on each occasion and the value considered for significance.
Results
N small, age with large variance, this can impact the results. How can authors minimize these biases?
Present the statistical tests made in the tables that need it.
Mark significant p values ​​with * to make the tables easier to read
Separate the results of the quanti and quali method into subsections. Present the analysis methodology of the quali method.
Discussion
It needs to bring applications to global levels of study. Weaknesses (N and age) need to be addressed or justified by previous studies.
Reviewer 3 Report
GENERAL COMMENTS
Dear authors,
Thank you for this article about physical activity barriers and facilitators on a cohort of 30 women.
The article is interesting, also considering the increasing importance of PA in contemporary society.
Despite this, I have some concerns about some sections and data in the manuscript.
Please see below for further comments.
INTRODUCTION
- I suggest better explain ExRx; for example, thus it contains the kind of physical activity, the intensity, the frequency, and repetitions for each suggested exercise or is it just a recommendation to practice PA?
- I also suggest considering the following article for the role of PA on obesity and eating disorders in general: Galasso et al., (2020) 10.3390/nu12123622.
MATERIALS AND METHODS
- I wonder if information in lines 93-111 is really necessary or if they could be delated or, instead, abbreviated.
- Line 133: could it be possible that this selection could have lead to bias or reduced sample size? If the authors had tried to recruit all the subjects visiting the centre, they could have had a larger sample. Could they explain why did they decide to contact only a small part of the women?
- Line 144: why did the authors opt for a telephone interview instead of a face-to-face or in-person interview? Maybe, the second one could have facilitated the enrollment.
- Lines 164-170: statistical analysis should be described more in-depth. Did the authors check the normality? Which are the alpha and the CI values?
- How did the authors decide that the cutoff values between low and high centre utilization should be 1?
RESULTS
- It is not clear if the final sample is composed of 20 or 26 women.
- Line 238: if some women never visited the centre, how could have they correctly respond to the questions? They could not have experience with the interpersonal or environmental levels.
- Line 239: could the free characteristic of the centre and the discount rate create a bias in the results? The centre fee could be one of the barriers to engaging in PA that is impossible to report in this condition.
- Table 1, employment: the difference is not significant.
- Table 1: CHAOS is not explained in the methods section.
- Line 256: in my opinion, the data referring to the education level merit more attention because usually the higher educated subjects visit the gym more frequently and for longer periods than the less educated ones.
- Line 273: <45 should be better specified.
- Line 280: in my opinion, it is better to say the exact number or percentage instead of often
DISCUSSION
- The authors may also consider articles referring to PA barriers in cancer patients or survivors. Even though the authors are focused on CVD and obesity disease, articles referring to cancer could offer more information and different points of view regarding barriers and facilitators for the PA practice.
Round 2
Reviewer 1 Report
The authors adequately answered the questions.